# Tuberculosis in Prisons: Importance of Considering the Clustering in the Analysis of Cross-Sectional Studies

**DOI:** 10.3390/ijerph20075423

**Published:** 2023-04-06

**Authors:** Diana Marín, Yoav Keynan, Shrikant I. Bangdiwala, Lucelly López, Zulma Vanessa Rueda

**Affiliations:** 1Facultad de Medicina, Universidad Pontificia Bolivariana, Medellín 050034, Colombia; 2Department of Medical Microbiology and Infectious Disease, University of Manitoba, Winnipeg, MB R3E 0J9, Canada; 3Department of Community Health Sciences, University of Manitoba, Winnipeg, MB R3E 0J9, Canada; 4Department of Internal Medicine, University of Manitoba, Winnipeg, MB R3E 0J9, Canada; 5Department of Health Research Methods, Evidence and Impact, McMaster University, Hamilton, ON L8S 4K1, Canada; 6Population Health Research Institute, McMaster University, Hamilton, ON L8L 2X2, Canada

**Keywords:** clustered-data, cross-sectional studies, log-binomial regression, modified Poisson regression, GEE, multilevel analysis, tuberculosis

## Abstract

The level of clustering and the adjustment by cluster-robust standard errors have yet to be widely considered and reported in cross-sectional studies of tuberculosis (TB) in prisons. In two cross-sectional studies of people deprived of liberty (PDL) in Medellin, we evaluated the impact of adjustment versus failure to adjust by clustering on prevalence ratio (PR) and 95% confidence interval (CI). We used log-binomial regression, Poisson regression, generalized estimating equations (GEE), and mixed-effects regression models. We used cluster-robust standard errors and bias-corrected standard errors. The odds ratio (OR) was 20% higher than the PR when the TB prevalence was >10% in at least one of the exposure factors. When there are three levels of clusters (city, prison, and courtyard), the cluster that had the strongest effect was the courtyard, and the 95% CI estimated with GEE and mixed-effect models were narrower than those estimated with Poisson and binomial models. Exposure factors lost their significance when we used bias-corrected standard errors due to the smaller number of clusters. Tuberculosis transmission dynamics in prisons dictate a strong cluster effect that needs to be considered and adjusted for. The omission of cluster structure and bias-corrected by the small number of clusters can lead to wrong inferences.

## 1. Introduction

People incarcerated in prisons have a higher incidence of tuberculosis (TB) disease and infection [1,2,3] compared to the general population [4]. Tuberculosis infection (TBI) and TB disease incidences vary between countries and cities, and the structural and social conditions of prisons differ between them. There are some specific characteristics that vary between prisons and between courtyard and cells inside each prison [5], such as overcrowding, poor ventilation, and limited access to health care services, which contribute to TB transmission [6,7,8,9]. In addition, there are individual factors that increase the risk of TBI acquisition and progression to TB disease, such as age, smoking, drug use, time of incarceration, history of prior TB, other underlying conditions (HIV, diabetes), extrapulmonary TB, place of residence, or immigration from high and medium TB-burdened countries, etc. [10,11,12,13,14,15,16].

People in prisons share similar environmental characteristics (exposures) because they are naturally grouped (confined) within cells, courtyards, and prisons. This is considered a natural cluster (Figure 1).

When researchers aim to study which factors are associated with TB in prisons using a cross-sectional study, clustering should be considered in the analysis. As part of our literature review to identify how many of the publications whose main objective was to estimate the prevalence of TB and the risk factors associated with TB in prisons considered the natural cluster structure in their analysis, we conducted a systematic search in PubMed using the following words: “Tuberculosis”, “Prisoners”, “Prison”, “Jails”, “Factors associated”, “Risk factors”, “Cross-sectional”, and “Prevalence”. Of the 354 papers we identified, 264 were excluded because they were not cross-sectional (See Appendix A). Among the 90 articles that reported cross-sectional studies, we found that only eleven adjusted by cluster (12.2%), five reported the prevalence ratio as the measure of association [5,17,18,19,20], and six reported the odds ratio [21,22,23,24,25,26]. The information about the search strategy and reviewed papers are reported in Appendix A.

Among observational studies, cross-sectional designs may involve cluster structures mainly of (1) people are grouped in a natural way (prisons, households, clinics, medical practices, and neighborhoods) and (2) of sampling designs, ranging from stratified random sampling to multi-stage sampling [27,28,29]. People who are grouped in a natural way tend to have more similar exposures regarding the event of interest, and therefore, failing to consider this structure in analyses leads to inappropriate statistical inferences [30,31,32].

Several modelling methods have been utilized to directly estimate the prevalence ratio when data have a cluster effect. Commonly used methods include log-binomial (log link) regression [33,34,35] and modified Poisson regression [36,37] with cluster-robust variance estimates (CRVE), log-binomial regression with Bayesian approach [38,39], mixed-effects regression models, and the marginal or population-averaged models using generalized estimating equations (GEE) with robust standard errors [29,40,41,42,43]. In all these methods, when data are clustered in more than one level, the most appropriate level must be chosen to estimate CRVE [44].

The number of clusters, the variability in cluster sizes and their impact on the estimation of measures effect have been studied mainly in randomized clinical trials [30,31,32,45], and the debate in cross-sectional studies has been focused on the choice of regression method [29,41,42,43,46,47] and the implications of the correlation matrix chosen [48] as well as regarding the most suitable measure of association [27,33,38,49,50,51].

Following the lessons learned from clinical trials and recognizing that clustering should be accounted for in the analysis, this study aims to evaluate the impact of either ignoring or adjusting for cluster structure and the selected level (prison, courtyard, and cell) using two cross-sectional datasets that studied TB disease and infection in prisons.

In this study we aimed to generate awareness among researchers investigating diseases in prisons regarding the presence of natural cluster structures and the importance of considering this factor in analyses.

## 2. Materials and Methods

To evaluate the impact of either ignoring or adjusting for cluster structure and the selected level, we used two cross-sectional datasets and four multivariable analyses: log-binomial, Poisson regression models, generalized estimating equations, and mixed-effects regression models with cluster-robust variance estimates. The studies are briefly described below:

### 2.1. Study 1: Active TB

The first corresponds to a cohort study of people deprived of liberty (PDL) with lower respiratory tract symptoms (*n* = 1305) that aimed to determine TB incidence and the factors associated with TB disease at baseline. Written informed consent forms were obtained for all participants. Full details about the study and procedures is available elsewhere [52].

#### 2.1.1. Variables

We took three variables that in the original study were associated with TB disease [43]: age ≤ 24 years (yes/no), history of previous TB (yes/no), and body mass index—BMI kg/m^2^ (normal: 18–25; low weight: <18; and overweight: >25). The outcome (Y) was TB disease (positive or negative).

#### 2.1.2. Sampling

Four prisons of medium- and maximum-security level in two cities were included (two male and two female prisons per city) [52]. We selected these prisons because Medellin has the highest burden of TB in Colombia and Bucaramanga has one of the highest TB incidences in the country.

Within these prisons, we recruited PDL in all 39 incarceration courtyards. The study has a hierarchical 4-level structure (city, prison, courtyard, PDL), each called a cluster, panel, or multilevel (Figure 1).

### 2.2. Study 2: TBI

The second study aimed to determine the risk factors and prevalence of TBI among PDL (*n* = 829). Written informed consent forms were obtained for all participants. Full details about the study and procedures are available elsewhere [5].

#### 2.2.1. Variables

We used the following variables: age in years (18–24/25–64/≥65), history of previous incarceration (yes/no), time of incarceration in the current prison in months (≤12/13–24/≥25), presence of BCG scar (yes/no), and last contact with a TB case in months (0/1–12/≥13). The outcome was LTBI (positive or negative).

#### 2.2.2. Sampling

Two male prisons in one city were selected, and individuals were randomly selected from 12 courtyards with high and lower incidences of TB [5]. There is a hierarchical 3-level structure (prison, courtyard, PDL).

We chose these prisons because both had a high TB incidence (>500/100,000 PDL).

### 2.3. Multivariable Regression Models

This section will present the most common multivariable methods used for modeling binary variables, such as having or not having TB. Additionally, the methods were used to determine when it is necessary or not necessary to adjust the cluster structure that is given when TB is studied in prisons. Figure 2 summarizes the regression models that we used to estimate prevalence ratio (PR) and odds ratio (OR) from independent or correlated data with the commands in STATA^®^ to adjust by cluster structure.

#### 2.3.1. Regression Models for Non-Clustered Dichotomous Outcomes

If we omit any cluster structure, the outcome of interest for each prisoner in study 1 (n = 1305) would be represented by Y_i_ (i = 1, 2, …, n), where Y_i_ = 1 means PDL with TB disease and Y_i_ = 0 if not.

In logistic regression, the multivariable regression model that reflects the relationship between independent variables and having TB disease is represented by Equation (1):(1)$$\begin{array}{ll} Logit\left(\pi_{i}\right)=\beta_{0} & +\beta_{1}{Age}_{\leq 24Years}+\beta_{2}Prior\;TB+\beta_{3}Low\;weight\\ & +\beta_{4}Overweight \end{array}$$
where π*_i_* is the probability that a subject *i* has TB disease [P(Y_i_ = 1)]. The reason that logistic regression estimates the OR is due to the logit transformation, which is called the link function between the independent variables and the Y_i_ outcome. This link function in log-binomial and Poisson regressions changes to log, and it is the reason for directly estimating PR instead of OR (Equations (2) (log-binomial) and (3) (Poisson)).
(2)$$\begin{array}{ll} Log\left(\pi_{i}\right)=\beta_{0}+ & \beta_{1}{Age}_{\leq 24Years}+\beta_{2}PriorTB+\beta_{3}Low\;weight\\ & +\beta_{4}Overweight \end{array}$$
(3)$$\begin{array}{ll} Log\left(\lambda_{i}\right)=\beta_{0}+ & \beta_{1}{Age}_{\leq 24Years}+\beta_{2}Prior\;TB+\beta_{3}Low\;weight\\ & +\beta_{4}Overweight \end{array}$$

In the Poisson regression model represented in Equation (3), *λ_i_* represents the mean of having TB, and the mean of dichotomous values (TB disease yes/no) is the same probability of having TB, *π_i_.* The Poisson model is a regression indicated for outcomes that one can “count” (for example, the number of transplants rejected and the number of revascularization procedures in patients with heart disease), and it has been suggested as an alternative method to binomial regression because it has associated convergence problems. Poisson regression estimates the consistent coefficients of Equation (3) but the variances are inconsistent. Variance in the Poisson regression is greater than the variance in the binomial regression unless the outcome is rare (prevalence <10%). To avoid overestimating standard errors for the estimated parameters, the robust variance estimator (sandwich variance estimator) is used in Poisson regression [36,53].

The confidence interval (CI) validity for the estimated PR or OR with any of the three regression methods must be corrected when the main assumption is violated, i.e., the observations are not independent. When people are recruited from places such as houses, neighborhoods, cities, prisons, and clinics, it is thought that the data have a clustered structure, implying that outcomes tend to be more similar within a cluster (i.e., they are correlated) and have greater variation between clusters. The quantification of this similarity or degree of correlation is estimated using the intraclass correlation coefficient (ICC), which is a measure that ranges between 0 and 1 [54]. The typical magnitude of an ICC is between 0.001 and 0.15 [31,36,37]. A strong cluster effect is considered when the ICC is above 0.15 [43].

#### 2.3.2. Multivariable Regression Models for Dichotomous Data with Clustered Structure

Clustered data can have multiple levels of grouping, and within each level, there are correlated observations. If we evaluate a person considering the three cluster levels, the binary outcome of interest would be written as Y_ijkl_. If Y_2312_ takes the value of 0, it indicates that person 2 of courtyard 1, prison 3, and city 2, does not have TB. When clusters are limited to the courtyard level, the outcome is expressed as Y_ijk_, and it becomes a count variable. The cluster structure is shown in Figure 1**.**

The simplest analytical approach to clustered data is to use any of the three previous regressions (logistic, log-binomial or Poisson). To adjust the CI to the cluster structure, one can use the modified sandwich variance estimator [37], cluster-robust standard errors, or cluster-robust variance estimates—CVRE. However, when there are multiple levels of clustering, the level for the CVRE must be chosen. It has been recommended to use the highest level whenever the ICC is less than 0.20 in models with multiple predictors [44].

Another approach for correlated data is to use marginal models or population-averaged models applying GEE. To estimate the association and its CI with GEE, it is necessary to specify the working correlation structure within the clusters. In a cross-sectional study, this correlation can be exchangeable (data within the cluster are correlated, and this correlation is the same in all clusters) or unstructured (the correlation is different in each cluster). In addition, to estimate the correct CI with GEE, the sandwich variance estimator is recommended due to being asymptotically robust regarding the misspecification of correlation structures [55,56].

In GEE, when the number of clusters is low (<40), the CIs tend to be narrower and the type I error rate is higher. Bias-corrected standard errors are required to improve the estimates, and their choice depends on the coefficient of variation (CV) and the number of clusters. The Kauermann and Carroll (KC) correction is suggested when the number of clusters is less than ten and the CV is <60%, and the Fay and Graubard (FG) when the CV is >60% [30,32,57].

A different approach when data are clustered, and perhaps the most widely used method in clustered RCT and etiological studies with hierarchical structures [29,41,55,58], are multilevel models with fixed or random effects. In cases where the interest of the researcher is focused on random effects, the variance components, missing data patterns, and the specific interpretations of the cluster should be considered and only multilevel analysis can be executed [31,58]. As with GEE, when a small number of clusters is used, the mixed models can lead to an inflated type I error rate and small-sample corrections are recommended for the analysis [30,31].

### 2.4. Analysis

We estimated TB prevalence for the total sample and each variable. To evaluate the changes in the measure of association, ignoring the cluster structure in study 1 and using different regression models, we estimated the adjusted OR and PR with the respective 95% CI using logistic regression, log-binomial regression, and robust Poisson regression. In addition, we compared the OR and the PR using the formula (((OR − PR)/PR) × 100), as reported by Martinez et al. [38]. This comparison is to analyze the effect of estimating an OR instead of a PR in a cross-sectional study.

Next, to evaluate the effect of the selected cluster level on the 95% CI, we used the GEE Poisson regression and the log-binomial and modified Poisson regression models with robust standard errors adjusted by each level (Courtyard = 39 clusters; Prison = 4 clusters and City = 2 clusters). In addition, we used the mixed-effects Poisson regression models considering the cluster structure. We implemented regression models with one clustering level, with more than one clustering level, and with levels of clustering as fixed effects. For each model, the ICC was estimated.

Finally, to evaluate the effect of the adjustment of cluster-robust standard errors and bias-corrected standard errors by courtyard cluster, in Study 2, we compared the PR and their CI obtained by the log-binomial, modified Poisson regression, Poisson mixed-effects model, and Poisson GEE model with exchangeable correlation. The measures of association and 95% CI obtained by models with or without adjustment were compared using Stata^®^ version 15.0 and R^®^ version 4.2.

## 3. Results

### 3.1. Effect of Prevalence in General and Sub-Groups Ignoring Clustering

Study 1 found a general prevalence of active tuberculosis of 5.52% (in 72 inmates) [43], with variations between 2.89% and 18%, according to the exposure factor in Table 1.

Comparing logistic regression (which estimates OR) to binomial regression or Poisson regression (which estimate PR) when the prevalence within the categories of exposure factors was greater than 10%, the OR was up to 25 times bigger than the PR. Robust Poisson regression and log-binomial regression displayed the same PR, with log-binomial regression yielding significantly narrower CIs. Table 1 shows the comparison.

### 3.2. Effect of Cluster Level on Cluster-Robust Variance Estimates

In Study 1, the number of people per courtyard varied between 1 and 284 (coefficient of variation—CV: 182% and intraclass correlation—ICC = 0.043) and per prison ranged between 36 and 796 (CV = 103% and ICC = 0.00).

When the cluster structure is considered in the analysis, the estimated measure of association is not affected by which cluster level is selected. However, the confidence interval width is wider if CRVE is adjusted for the lowest level (courtyard), meaning that the ICC and, thus, the design effect, is larger at the more proximal level. When the four regressions are compared at the lowest level, it can be observed in Table 2 that the CIs estimated with GEE and mixed models are narrower than the other two methods.

The adjustment of the cluster effect can uncover differences, and CI can become significant compared to CI without adjustment by cluster effect. This was seen in the case of age >24, where the association became significant after the adjustment of any cluster level (Table 2 versus Table 1).

In multilevel analysis, it is necessary to define whether the objective is only to adjust for the correlation or to understand how much of the correlation is due to each clustering level. In the first scenario, it is recommended to employ fixed effects, and in this situation, model number 4 from Table 3 could be analyzed. In case the researcher is not interested in evaluating the effect of each prison, random effects are recommended; in this case, model 3 from Table 3 might be used because the ICC values for city and prison show a negligible effect on TB incidence.

### 3.3. Effect of Adjustment by Cluster-Robust Standard Errors and Bias-Corrected Standard Errors

Table 4 shows the changes in the CI between the models with and without adjustment by cluster-robust standard errors and number of clusters. Without any correction, the CIs are broader because the cluster structure has an influence (ICC = 0.07). After adjustment by cluster structure, a number of factors became significant (age, prior incarceration, and time of prior incarceration). However, with the additional adjustment by the number of clusters using the Fay and Graubard correction (bias-corrected standard errors), the significance disappears and the confidence intervals (CIs) become even wider.

## 4. Discussion

As we show in this study, in the context of tuberculosis in prisons, the effect of clustering is an important consideration, and therefore should be considered in the analysis. We found that the association between exposure and outcome might be missed or changed due to the omission of the cluster structure in the model. In addition, failure to adjust the correct clustering level directly influences the reliability of the results. The latter illustrates that choosing the best model should be based on the number of clusters of each level in the coefficient of variation and the ICC.

We found that when analysis ignored the cluster structure, the associations and CI estimated by Poisson and log-binomial regression was similar, with CIs being slightly narrower using Poisson regression [37]. In addition, when we compared the crude and adjusted measures, the measures of associations changed in the three models, and the precision improved when other variables were considered. This finding contrasts with that of Zou GY [37], who suggested that one of the advantages of PR is the little to no variation after adjustment for other variables, in contrast to the effect on the OR.

Differences in the measures of association between regression models that consider the cluster structure and those obtained by standard regression models without accounting for cluster structure have been reported [43,59,60]. There are higher standard errors (confidence intervals broader) [42,59,61] and higher rates of the type I error [59,60]. Our study found that when we compared the four regressions at the lowest level (courtyard), the CIs obtained from GEE and mixed models were narrower than the other two methods.

We demonstrate herein that cluster effects influence the results that estimate the risk factors associated with TB disease and TB infection in prisons and that clustering needs to be considered in the analysis. Tuberculosis is communicable and transmitted via the respiratory route. The transmission efficiency depends on factors such as crowding, time spent with an index case, and air circulation. Cluster structure governs attributed variables that are external to individuals. These variables explain the differences in the outcomes between cities, prisons, or courtyards, in contrast to some examples of non-transmissible conditions, such as growth stunting, as studied by Bhowmik [42]. Hence, the biology of the disease or agent studied must be considered when selecting the analysis method or interpreting the impact of clustering.

The adjusted prevalence ratio estimation to acknowledge that the outcome is correlated with cluster structure has been modelled primarily with marginal models through generalized estimation equations (GEE) and mixed models [40,42,43]. In addition, researchers have compared the efficiency of the models that incorporate the cluster-robust standard errors using mixed models to estimate the PR [46]; modified Poisson regression versus log-binomial [36,37,62] and Cox regression [62]; mixed-effects logistic regression versus mixed-effect Poisson regression [61]; and mixed-effects logistic regression using the conditional and marginal standardization methods RP [61]. The published literature indicates that the comparison between all these regression models should focus on the ability to provide more precise CIs because it is well-known that they will yield a different measure of association [43,59,60].

Under this premise, there is no consensus so far on when to use GEE or a mixed model in this context, as some report more precise standard errors (narrower CI) obtained with GEE [36,40,43] when the robust standard errors [32,40,55] and independent of the specified correlation matrix are used [32,36,40,42]. In contrast, others have shown the superiority of mixed models compared with GEE and survey logistic regression (SLR). Bhowmik [42] found that among the models that consider the clustered structure, the random intercept logistic regression model resulted in the lowest standard error and the highest correct percentage of classification compared to SLR and GEE. In our study, when disease prevalence was less than 10%, the CI from GEE was narrower than the mixed-effects regression regardless of the number of clusters.

Our study used modified Poisson and log-binomial regression to estimate the PR with its CI adjusted by cluster directly. When the disease prevalence was less than 10%, the two methods identified similar risk factors regardless of the number of clusters, but the confidence intervals were narrower with modified Poisson regression.

Other studies that compared Poisson regression performance with log-binomial regression [36,37,53,62] highlighted the superiority of the Poisson regression due to the problems of convergence [36,37,53,62] and bias [36] associated with the log-binomial regression. Additionally, these studies found that the methods produce similar type I error rates [36], similar PR estimates, and narrower confidence intervals with Poisson, irrespective of the number of clusters, with further improvement with the interchangeable correlation matrix [37].

The effect that the number and size of the clusters have on the estimates has been an area of debate in the literature. Most comparisons arise from clustered randomized clinical trial studies examining the effect that the number and size of clusters have on the results obtained when analyzing the data. Studies with a small number of clusters [30] and high heterogeneity in cluster sizes [31] tend to inflate the rate of type I error and decrease the power when they are analyzed using standard methods. Some corrections for cluster size and the number of clusters are related to the intraclass-correlation coefficient (ICC) and the heterogeneity of cluster size measured through the coefficient of variation (CV).

The GEE method with robust standard errors underestimates the variance and obtains inflated type I error rates when the number of clusters is less than 40 [32] and is not recommended when there are fewer than eight clusters [30,40]. Despite the preceding, Leyrat showed that GEE with robust standard errors and small-sample corrections to correct the standard error has similar results to the mixed models in continuous outcomes [30]. Li [32] recommends using the correction of Kauermann and Carroll (KC) when the number of clusters is less than ten and the CV is less than 60% and, when the CV is >60%, suggests using the correction of Fay and Graubard (FG).

In mixed models, it has been shown that when the number of clusters is below 30, these models inaccurately estimate standard errors and low variance components [41,43] and biased estimates (greater possibility of a type I error) [41] and are associated with worse power [31,41]. In those mixed models with two levels and few clusters at level 2, the problems arise mainly when estimating standard coefficients and errors skewed at that level 2 [29,41,63]. This is attributed to the number of participants (level 1) which are frequently greater than the number of clusters (Level 2), resulting in an insufficient sample size to obtain valid estimates at that level. A minimum of 30 clusters is required to obtain valid type I error rates [31] and minimize skewed estimates. Leyrat and Li [30,31] indicate that the higher the cluster size variation and the lower the number of clusters, the greater type I error rate and worse statistical power. They also suggest using the between–within (B-W) degrees of freedom approximation method for mixed models with dichotomous outcomes, a small number of clusters (<30, even as low as 10) and a large variation of cluster size. However, there are a few programs that incorporate these corrections. We reviewed the papers that used those corrections compared to uncorrected estimations. There are differences only in the third decimal of the measure of associations and their confidence intervals, which would not have clinical or practical implications.

There is no consensus on which cluster level to use to adjust the CRVE. However, it is recommended to adjust the analyzes by clustering when it exists.

## 5. Conclusions

Using real cohort data instead of simulations, we aimed to understand the cluster effect when researchers evaluate potential risk factors associated with TB disease and TB infection in prisons. When researchers, epidemiologists, or statisticians analyze research conducted in settings with clear physical clustering, such as prisons, cluster structures need to be considered. The number and the type of clusters need to be adjusted for, as these impact the measures of association and their confidence intervals compared to standard methods. Each choice of the four regression models to adjust by cluster has advantages and disadvantages. The selection of the cluster level has to be driven by the knowledge of the biology and epidemiology of the studied condition, as criteria for choosing the city, prison, or courtyard depend on the heterogeneity of the problem studied. In addition, the model selection depends on the knowledge of the statistical method used, the software available for analysis, and the types of variables, as some variables affect the convergence of the model. Independent of the model chosen, all four models must be adjusted for cluster-robust standard errors.

Here, we demonstrated the relevance of the studied disease model and the critical impact of the cluster size, structure, and methods for adjusting the estimations. Adjustment by cluster critically affects precision and should, therefore, be reported in cross-sectional studies to improve reproducibility and transparency.

## Figures and Tables

**Figure 1 ijerph-20-05423-f001:**
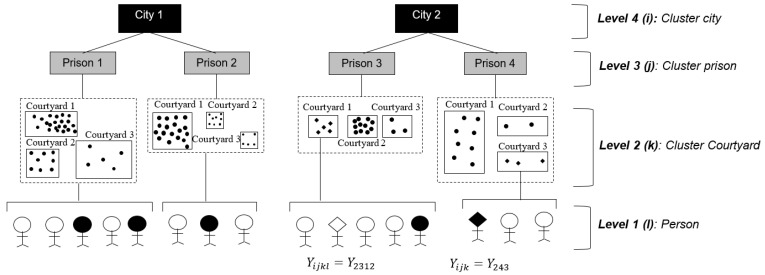
Multilevel structure used to determine factors associated with TB disease in an analytical cross-sectional study. Legend: In Level 1, a black circle represents a person with TB disease and a white circle represents an individual without TB. In Level 2, the population density within each courtyard is represented by the number of dots; a greater number of dots indicates a higher population density inside the courtyard. People with a black rhombus illustrate the coding system for the outcome at different levels. In courtyard 1 of prison 2, the fact that two or more people have TB disease inside that courtyard can be attributed to shared conditions, such as population density and overcrowding, which facilitate the emergence of the disease (intra-cluster variation); these conditions may differ from the conditions of people who have TB disease in courtyard 3 (inter-cluster variation).

**Figure 2 ijerph-20-05423-f002:**
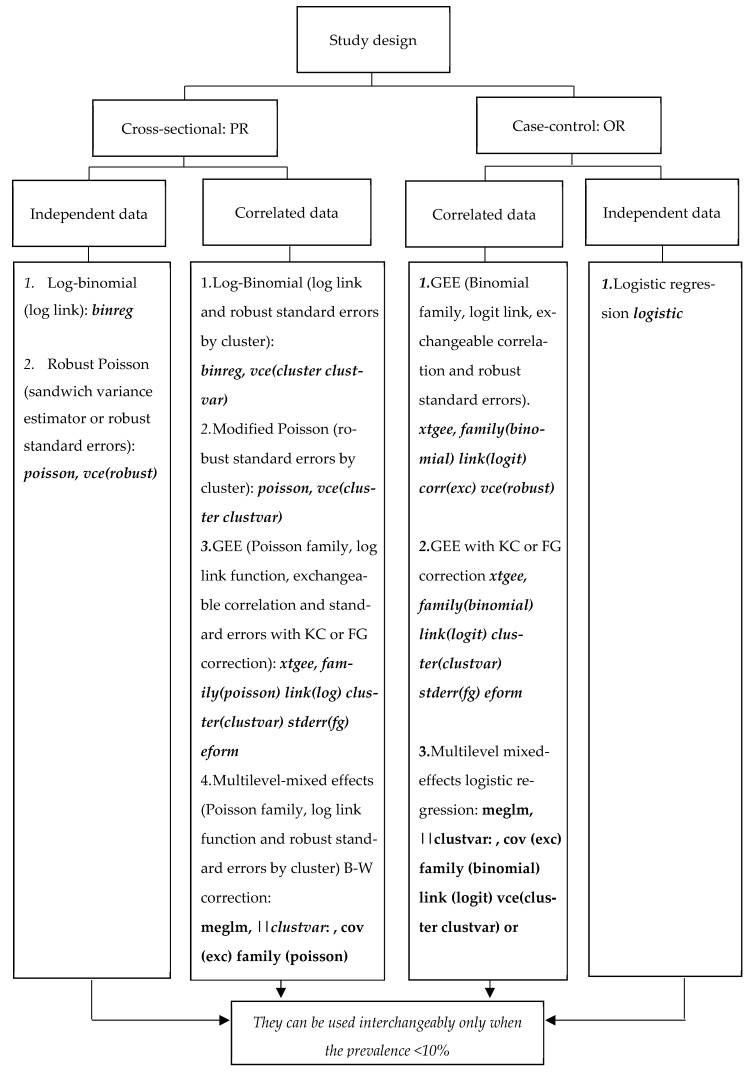
Diagram to select regression models that directly estimate prevalence ratio (PR) or odds ratio (OR) from independent or correlated data. Legend: In GEE, Li [31] recommends using the correction of Kauermann and Carroll (KC) when the number of clusters is less than ten and the coefficient of variation (CV) is less than 60%. When the CV is >60%, the use of the correction outlined by Fay and Graubard (FG) has been suggested. Leyrat and Li [30,31] recommend using the between–within (B-W) degrees of freedom approximation method in mixed models for binary or continuous outcomes and a low number of clusters (<30). In bold are the relevant Stata commands. Abbreviations: GEE, generalized estimation equations; KC, correction of Kauermann and Carroll; FG, correction Fay and Graubard; B-W, approximation of degrees of freedom between–within, according to the number of clusters and coefficient of variation.

**Table 1 ijerph-20-05423-t001:** Comparison of the odds ratio (OR) and prevalence ratio (PR) estimated with three types of regressions when the active TB prevalence is less than 10% and under the assumption of independent data.

Exposure Factor	Active TB Prevalence	Logistic	Log-Binomial ^1^	Robust Poisson ^2^	Comparison ^3^
		OR_adjusted_	PR_adjusted_	PR_adjusted_	
		[95%CI]	[95%CI]	[95%CI]	
Age ≤ 24 (ref)	4.61				
Age > 24	5.84	1.43	1.40	1.39	7.5
		[0.797, 2.547]	[0.809, 2.403]	[0.798, 2.405]	
No prior TB (ref)	5.02				
Prior TB	18.0	3.31	2.84	2.84	25.5
		[1.461, 7.504]	[1.469, 5.494]	[1.447, 5.569]	
Normal BMI (ref)	5.36				
Low weight	16.0	3.32	2.93	2.89	20.2
		[1.677, 6.576]	[1.660, 5.163]	[1.638, 5.096]	
Overweight	2.89	0.42	0.43	0.43	1.8
		[0.147, 1.169]	[0.158, 1.187]	[0.156, 1.185]	

Abbreviations: BMI, body mass index; CI, confidence interval; OR, odds ratio; PR, prevalence ratio; ref, reference category; TB, tuberculosis; body mass index: low weight < 18 kg/m^2^ and overweight > 25 kg/m^2^. OR_adjusted_ and PR_adjusted_: multivariable models adjusted for age, prior TB, and BMI. ^1^. Generalized linear model, binomial family, log link function. Estimation method: maximum likelihood (ML). ^2^. Robust standard errors: sandwich variance estimator. ^3^. Comparison = ((OR − PRLog-binomial)/PRLog-binomial) × 100. The measures of association obtained by Poisson regression and log-binomial regression were identical. Consequently, we only compared OR and PR using log-binomial and logistic regression.

**Table 2 ijerph-20-05423-t002:** Comparison of the prevalence ratio of TB disease with four regressions for correlated data when the overall prevalence of the disease is less than 10% and is considered different cluster levels to adjust the robust variance error.

Exposure Factor	Adjusted Prevalence Ratio and [95%CI]											
	GEE ^1^			Multilevel—Mixed-Effects ^2^			Log-Binomial ^3^			Modified Poisson ^4^		
Level	Courtyard (n = 39)	Prison (n = 4)	City (n = 2)	Courtyard (n = 39)	Prison (n = 4)	City (n = 2)	Courtyard (n = 39)	Prison (n = 4)	City (n = 2)	Courtyard (n = 39)	Prison (n =4 )	City (n = 2)
Age ≤ 24 (ref)												
Age > 24	1.38	1.37	1.37	1.38	1.38	1.38	1.40	1.40	1.40	1.39	1.39	1.39
	[1.020, 1.874]	[1.089, 1.729]	[1.310, 1.433]	[1.008, 1.878]	[1.020, 1.856]	[1.202, 1.575]	[1.015, 1.918]	[1.056, 1.843]	[1.256, 1.549	[1.011, 1.898]	[1.043, 1.841]	[1.245, 1.542]
No prior (ref)												
Prior TB	2.73	2.82	2.81	2.67	2.67	2.67	2.84	2.84	2.84	2.84	2.84	2.84
	[1.531, 4.852	[1.944, 4.075]	[1.988, 3.962]	[1.542, 4.637]	[1.644, 4.349]	[1.529, 4.676]	[1.571, 5.140]	[1.902, 4.246]	[1.886, 4.283]	[1.645, 4.901]	[1.757, 4.587]	[1.669, 4.828]
Normal BMI (ref)												
Low weight	2.85	2.85	2.84	2.84	2.84	2.84	2.93	2.93	2.93	2.89	2.89	2.89
	[1.842, 4.408	[1.695, 4.781]	[2.211, 3.639]	[1.786, 4.514]	[1.615, 4.994]	[2.116, 3.811]	[1.885, 4.545]	[1.628, 5.264]	[2.048, 4.185]	[1.814, 4.601]	[1.624, 5.138]	[2.123, 3.932]
Overweight	0.42	0.43	0.44	0.42	0.42	0.42	0.44	0.44	0.44	0.43	0.43	0.43
	[0.119, 1.474	[0.059, 3.116]	[0.049, 3.863]	[0.119, 1.491]	[0.035, 5.027]	[0.158, 11.281]	[0.125, 1.503]	[0.038, 4.896]	[0.017, 10.869]	[0.123, 1.501]	[0.037, 4.877]	[0.017, 10.846]
CV(%)	181%	103%	-									
ICC				0.043	<0.001	<0.001						

Abbreviations: BMI, body mass index; CI, confidence interval; GEE, generalized estimation equations; ref, reference category; TB, tuberculosis; CV, coefficient of variation; ICC, intraclass correlation. Body mass index: low weight < 18 kg/m^2^ and overweight > 25 kg/m^2^. ^1^. Poisson family, log link function. Robust standard errors. Exchangeable correlation. ^2^. Multilevel mixed-effects Poisson regression had the exposure factors as fixed effects and the city, the prison, and the courtyard as random effects. The city model is a four-level model (people deprived of liberty (PDL, level 1) clustered within courtyards (level 2), clustered within prisons (level 3), and clustered within cities). The prison model is a three-level model, and the courtyard is a two-level model (PDL clustered within courtyards). Cluster-robust standard errors are adjusted by the highest level in each multilevel model. The ICC is shown for the highest level in each multilevel model. ^3^. Generalized linear model, binomial family, log link function. Cluster-robust standard errors. ^4^. Cluster-robust standard errors.

**Table 3 ijerph-20-05423-t003:** Comparison of multilevel models for estimating TB prevalence utilizing the city, prison, and courtyard levels as fixed or random effects.

Exposure Factor	Model 1: City ^1^		Model 2: Prison ^1^		Model 3: Courtyard ^1^		Model 4 ^2^		Model 5 ^2^	
Level	PR	[95%CI]	PR	[95%CI]	PR	[95%CI]	PR	[95%CI]	PR	[95%CI]
Age ≤ 24 (ref)										
Age > 24	1.39	[1.25–1.54]	1.39	[1.04–1.84]	1.38	[1.01–1.88]	1.39	[1.02–1.86]	1.39	[1.02–1.88]
No prior (ref)										
Prior TB	2.84	[1.67–4.83]	2.84	[1.76–4.59]	2.67	[1.54–4.64]	2.70	[1.58–4.60]	2.68	[1.55–4.62]
Normal BMI (ref)										
Low weight	2.89	[2.12–3.93]	2.89	[1.62–5.14]	2.84	[1.79–4.51]	2.79	[1.75–4.45]	2.84	[1.78–4.53]
Overweight	0.43	[0.02–10.84]	0.43	[0.04–4.88]	0.42	[0.12–1.49]	0.42	[0.12–1.51]	0.42	[0.12–1.49]
Prison D (ref) ^3^										
Prison A	-	-	-	-	-	-	1.64	[0.90–2.99]	-	-
Prison B	-	-	-	-	-	-	1.63	[0.76–3.47]	-	-
Prison C	-	-	-	-	-	-	2.15	[1.02–4.54]	-	-
City A (ref) ^3^										
City B	-	-	-	-	-	-	-	-	1.42	[0.79–2.54]
ICC	<0.001		<0.001		0.043		0.031		0.034	

Abbreviations: BMI, body mass index; PR, prevalence ratio; 95%CI, 95% confidence interval; GEE, generalized estimation equations; ref, reference category; TB, tuberculosis; ICC, intraclass correlation. Body mass index: low weight < 18 kg/m^2^ and overweight > 25 kg/m^2^. ^1^.Multilevel mixed-effects Poisson regression with a single random effect: model 1 = city, model 2 = prison, and model 3: courtyard. ^2^.Multilevel mixed-effect Poisson regressions with the courtyard as a random effect and prison or city as fixed effects. Cluster-robust standard errors adjusted by the courtyard factor. ^3^.The reference category was those with the lowest TB incidence.

**Table 4 ijerph-20-05423-t004:** Comparison of the prevalence ratio and 95% confidence intervals associated with latent tuberculosis infection without and with adjustment by cluster-robust standard errors and number of clusters.

Exposure Factor	GEE ^1^		
	Without Any Adjustment ^3^ [95%CI]	With CVRE [95%CI]	With BCSE [95%CI]
Age 18–24 (ref) (Years)			
25–64	1.14 [0.892, 1.407]	1.14 [1.087, 1.193]	1.14 [0.658, 1.970]
≥65	1.10 [0.687, 1.473]	1.10 [0.996, 1.221]	1.10 [0.495, 2.457]
History of the previous incarceration	**1.09 [0.904, 1.311]**	**1.09 [1.009, 1.172]**	**1.09 [0.650, 1.819]**
Time of current incarceration ≤12 (ref) (Months)			
13–24	**1.11 [0.878, 1.408]**	**1.12 [1.007, 1.228]**	**1.12 [0.562, 2.201]**
≥25	1.08 [0.874, 1.332]	1.08 [0.951, 1.234]	1.08 [0.392, 2.995]
Presence of BCG scar	1.00 [0.931, 1.070]	1.00 [0.985, 1.013]	1.00 [0.832, 1.198]
Last contact with a TB case—No contact (ref ) (Months)			
1–12	0.98 [0.776, 1.235]	0.99 [0.862, 1.134]	0.99 [0.357, 2.740]
≥13	1.04 [0.797, 1.383]	1.04 [0.983, 1.107]	1.04 [0.689, 1.578]
Coefficient of variation	191%		
**Exposure Factor**	**Multilevel– Mixed-Effects ^2^**		
	**Without Any Adjustment [95%CI]**	**With CVRE [95%CI]**	**With B-W ^4^**
Age 18-24 (ref) (Years)			
25–64	**1.09 [0.877, 1.364]**	**1.09 [1.023, 1.169]**	-
≥65	**1.05 [0.724, 1.509]**	**1.05 [0.963, 1.134]**	-
History of the previous incarceration	**1.09 [0.904, 1.304]**	**1.09 [1.010, 1.168]**	-
Time of current incarceration ≤12 (ref) (Months)			
13–24	1.09 [0.865, 1.372]	1.09 [0.994, 1.193]	-
≥25	1.02 [0.836, 1.252]	1.02 [0.871, 1.203]	-
Presence of BCG scar	1.00 [0.936, 1.072]	1.00 [0.993, 1.011]	-
Last contact with a TB case—No contact (ref) (Months)			
1–12	1.00 [0.804, 1.262]	1.01 [0.862 1.176]	-
≥13	1.02 [0.777, 1.347]	1.02 [0.955, 1.095]	-
Intraclass Correlation	0.071		

Abbreviations: BCG, Bacillus Calmette–Guérin; CI, confidence interval; GEE, generalized estimation equations; TB, tuberculosis; CVRE, cluster-robust variance estimates; BCSE, bias-corrected standard errors by Fay and Graubard. Bold numbers represent a change in the association between variables and latent tuberculosis infection after adjustment.^1^ Poisson family, log link function. Exchangeable correlation. Cluster-robust standard errors adjusted by courtyard (n = 10). ^2^. Mixed-effects Poisson regression model. Exchangeable covariance. Cluster-robust standard errors (adjusted for 10 clusters in the courtyard). ^3^ There is no adjustment of standard errors by cluster structure and number of clusters. The CVRE is applied first, followed by the BCSE or B-W. ^4^. Between–within degrees of freedom approximation for dichotomous outcome are not available in Stata or R.

## Data Availability

The data presented in this study are openly available in https://doi.org/10.6084/m9.figshare.21751436.v2 (accessed on 15 March 2023).

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
