# Peer review of "Tuberculosis in Prisons: Importance of Considering the Clustering in the Analysis of Cross-Sectional Studies"

_ijerph, 2023, doi:10.3390/ijerph20075423_

Round 1

Reviewer 1 Report

Report on “Tuberculosis in prisons. Importance of considering the clustering in the analysis in cross-sectional studies”
In this manuscript, the authors evaluated the impact of adjustment versus failure to adjust by clustering on prevalence ratio (PR) and 95% Confidence Interval (CI) by using two cross-sectional datasets and four multivariable analyses. The findings showed that cluster structures need to be considered when researches studied Tuberculosis transmission dynamics in prisons. The omission of cluster structure and bias-corrected by the small number of clusters can lead to wrong inferences. The results are interesting. But there are some problems that need to be improved.
1. In Abstract, “OR” should be “odds ratio (OR)”.
2. In Introduction, the first paragraph is too short with only one sentence.
Please give more details that statistical methods used in TB transmission.
3. I suggested that the authors provided a flow chart for the systematic literature review in addition to Table S1.
4. In line 114, “100.000” should be “100,000”. Line 123, “24 anos” should be “24 years”.
5. Please put equation (1) into one line and use the same format with initial in capital like “Overweight”.
6. What are the advantage and disadvantage of the multivariable analyse? And what are the shortages of this study?

Author Response

March 15, 2023

Dear

Academic Editor

Tuberculosis (TB) Prevention and Care: A Global Public Health Issue

Thanks so much to the reviewers for their suggestions. All of them were very useful to improve our paper. Please find below the answer to each of their suggestions, and the manuscript with track-changes.

Academic Editor

This is an interesting manuscript which is very relevant to our current issue. Minor points should be addressed before publication (please refer to reviewers' comments). Importantly, it is central that the introduction of the manuscript is improved: The first paragraph is irrelevant and should not be the opening line to the manuscript. I would recommend starting with paragraph 3, developing the background on TB transmission, prisons and then build the case for clusters and the methods that deal with this, to finally set an aim or hypothesis to guide the reader further.

Answer: Dear editor thanks for your comment. We addressed your and all reviewers’ comments.

We changed the introduction following your suggestion.

Reviewer 1:

  1. Report on “Tuberculosis in prisons. Importance of considering the clustering in the analysis in cross-sectional studies” In this manuscript, the authors evaluated the impact of adjustment versus failure to adjust by clustering on prevalence ratio (PR) and 95% Confidence Interval (CI) by using two cross-sectional datasets and four multivariable analyses. The findings showed that cluster structures need to be considered when researchers studied Tuberculosis transmission dynamics in prisons. The omission of cluster structure and bias-corrected by the small number of clusters can lead to wrong inferences. The results are interesting.

Answer: Dear reviewer thanks so much for kind comments.

  1. But there are some problems that need to be improved. In Abstract, “OR” should be “odds ratio (OR)”.

Answer: Thanks, we defined the OR.

  1. In Introduction, the first paragraph is too short with only one sentence.
    Please give more details that statistical methods used in TB transmission.

Answer: Thanks for your comment. The three reviewers and the editor suggested to change the introduction. We changed it to make it clearer, focused on our main objective and to improve the reading.

  1. I suggested that the authors provided a flow chart for the systematic literature review in addition to Table S1.

Answer: Thanks, the flow chart was included as the supplementary Figure S1.

  1. In line 114, “100.000” should be “100,000”. Line 123, “24 anos” should be “24 years”.

Answer: You are right, thanks. We corrected these typos.

  1. Please put equation (1) into one line and use the same format with initial in capital like “Overweight”.

Answer: Thanks, we capitalized each word.

  1. What are the advantage and disadvantage of the multivariable analyse? And what are the shortages of this study?

Answer: Thanks for these questions. This manuscript aims ‘evaluate the impact of either ignoring or adjusting for cluster structure and the selected level (prison, courtyard and cell)’ in studies that have clustered data. The ‘advantage’ of using a multivariable model that accounts for the clustering in the data is that it is the correct model to use; failure to use the proper model (i.e. using a model that ignores the clustering structure in the data) leads to lower variances, inflated effect sizes, and possible erroneous interpretation of the statistical significance of the results. The ‘disadvantage’ of using a multivariable model that accounts for the clustering in the data is that researchers may not be aware of how to fit and interpret the models. Thus, the goal of this manuscript is to generate awareness among researchers in the fitting and interpretation of these models.

We chose to use data from two real studies rather than artificial data from simulations to illustrate the impact of different modeling approaches for analyzing studies that have clustered data. The advantages of using real data from our study of TB in prisons are that we contextualized the illustration of the methodological issues, and provided meaningful ways to interpret the results from these models.

The limitations of using our study of TB in prisons are that we are limited by the actual sample size and hierarchical structure of the dataset – a 4-level multivariate cluster design (2 cities, 4 prisons, 39 incarceration courtyards, and 1305 PDLs) in Study 1, and a 3-level multivariate cluster design (2 prisons in one city, 12 incarceration courtyards, and 829 PDLs) in Study 2. Future studies in different places may have higher number of clusters because of bigger prisons with more courtyards, cells and people, which it will not change the conclusions of our study regarding the need to adjust by cluster effect. What future researchers will need is to decide at which level they have to adjust their multivariable analysis.

Sincerely,

Diana Marín

Reviewer 2 Report

1. Title sould be better clarified as where the prisons are located in.

2. Abstract ok

3. The opening of the introduction makes no sense, and it should be properly modified.

4. In the part of introduction (line 61-67), this is more systematic review, and this study was designed as cross-sectional study.

5. 2.1.2 How are these 4 prisons selected into the research?

6. 2.3 It is ok to present the statistical methods, but it should be more understandable.

7. Most of the figures are ok.  The table 4 can be improved, synthesised or split into 2.

8. References are way too old.  They should be up-to-date.

9. English is ok and with enough scientific soundness.

10. The overall design of this study is confusing.  I do not concent the core concept of the study by the expression of this paper.

Author Response

March 15, 2023

Dear

Academic Editor

Tuberculosis (TB) Prevention and Care: A Global Public Health Issue

Thanks so much to the reviewers for their suggestions. All of them were very useful to improve our paper. Please find below the answer to each of their suggestions, and the manuscript with track-changes.

Academic Editor

This is an interesting manuscript which is very relevant to our current issue. Minor points should be addressed before publication (please refer to reviewers' comments). Importantly, it is central that the introduction of the manuscript is improved: The first paragraph is irrelevant and should not be the opening line to the manuscript. I would recommend starting with paragraph 3, developing the background on TB transmission, prisons and then build the case for clusters and the methods that deal with this, to finally set an aim or hypothesis to guide the reader further.

Answer: Dear editor thanks for your comment. We addressed your and all reviewers’ comments.

We changed the introduction following your suggestion.

Reviewer 2

  1. Title should be better clarified as where the prisons are located in.

Answer: Dear reviewer we understand why you suggest this comment. However, the focus of this paper is to evaluate the impact of either ignoring or adjusting for cluster structure and the selected level (prison, courtyard and cell) using two cross-sectional datasets when studying TB in prisons, no matter the location. Our paper illustrates the statistical implications of not adjusting by cluster, as well as, omitting adjustment for the low number of clusters.

  1. Abstract ok

Answer: Thanks for the comment.

  1. The opening of the introduction makes no sense, and it should be properly modified.

Answer: Thanks for the suggestion. As the three reviewers and editor suggested to modify the introduction, we changed it to make it clearer and to add some new references.

  1. In the part of introduction (line 61-67), this is more systematic review, and this study was designed as cross-sectional study.

Answer: We did a systematic search as part of our literature review but not a systemic review.

The search was systematic to identify how many of the publications whose main objective was to estimate the prevalence of TB and the risk factors associated with TB in prisons considered the natural cluster structure in their analysis. The fact that only 12% of the published papers adjusted by cluster, shows how many researchers are ignoring or are not aware about the cluster structure in prisons. This finding also highlights the importance of our paper.

  1. 1.2 How are these 4 prisons selected into the research?

Answer: We selected these prisons because Medellin has the highest burden of TB in Colombia, and Bucaramanga has one of the highest TB incidences in the country. We included this information in the methods section.

  1. 3 It is ok to present the statistical methods, but it should be more understandable.

Answer: Thanks. We tried to improve the presentation of the methods to make it more understandable.

  1. Most of the figures are ok.  The table 4 can be improved, synthesized or split into 2.

Answer:  Thanks for the comment. Table 4 was split into two sections.

  1. References are way too old.  They should be up-to-date.

Answer: Thanks. The references were updated in the introduction.

  1. English is ok and with enough scientific soundness.

Answer: Thanks for your comment.

  1. The overall design of this study is confusing.  I do not consent the core concept of the study by the expression of this paper.

Answer: Thanks for your comment. We changed the Introduction and modified extensively the Methods section to clarify the design of our study and our aim. We also wrote clearly the aim of our study: “to evaluate the impact of either ignoring or adjusting for cluster structure and the selected level (prison, courtyard and cell) using two cross-sectional dataset that studied TB disease and infection in prisons”.

Sincerely,

Diana Marín

Reviewer 3 Report

The paper concerns about the model of TBC clusters in prison, using  log-binomial regression and  Poisson regression. As reported, considering a cluster structure in the model of TBC spreading is useful to improve the infections controls.

Please consider to improve your bibliography by citing  doi:10.3390/healthcare10020386 in the introduction. I advise you to cite this and other related paper at line 39-40 in order to stress the importance of the epidemiological governance of infection disease among prisoners. TBC is spreading with cluster structure because of the increase of immigration, poverty and overcrowded prisons.  

Please, explain better why did you do a systematic review. At the first reading, it's seem that you analyzed a cluster of your population; it is not clearly that you analyzed other studies considering the cluster model of these.  

Results are well presented, the article is well written, I have only a little concerns of the real impact of the using this cluster model on all cross sectional studies. 

I have no more comments, good work.

Author Response

March 15, 2023

Dear

Academic Editor

Tuberculosis (TB) Prevention and Care: A Global Public Health Issue

Thanks so much to the reviewers for their suggestions. All of them were very useful to improve our paper. Please find below the answer to each of their suggestions, and the manuscript with track-changes.

Academic Editor

This is an interesting manuscript which is very relevant to our current issue. Minor points should be addressed before publication (please refer to reviewers' comments). Importantly, it is central that the introduction of the manuscript is improved: The first paragraph is irrelevant and should not be the opening line to the manuscript. I would recommend starting with paragraph 3, developing the background on TB transmission, prisons and then build the case for clusters and the methods that deal with this, to finally set an aim or hypothesis to guide the reader further.

Answer: Dear editor thanks for your comment. We addressed your and all reviewers’ comments.

We changed the introduction following your suggestion.

Reviewer 3

  1. The paper concerns about the model of TBC clusters in prison, using log-binomial regression and Poisson regression. As reported, considering a cluster structure in the model of TBC spreading is useful to improve the infections controls.

Answer: Dear reviewer thanks so much for your comment.

  1. Please consider to improve your bibliography by citing doi:10.3390/healthcare10020386 in the introduction. I advise you to cite this and other related paper at line 39-40 in order to stress the importance of the epidemiological governance of infection disease among prisoners. TBC is spreading with cluster structure because of the increase of immigration, poverty and overcrowded prisons.

Answer: Thanks. As all three reviewers and the editor suggested to modify the introduction, we changed it to improve the reading and make it clearer. We also included the suggested reference.

  1. Please, explain better why did you do a systematic review. At the first reading, it's seem that you analyzed a cluster of your population; it is not clearly that you analyzed other studies considering the cluster model of these.  

Answer: Two of the reviewers were confused by our systematic search. We did a systematic search as part of our literature review but not a systemic review. The search was systematic to identify how many of the publications whose main objective was to estimate the prevalence of TB and the risk factors associated with TB in prisons considered the natural cluster structure in their analysis. The fact that only 12% of the published papers adjusted by cluster structure, shows how many researchers are ignoring or are not aware about the cluster structure in prisons. This finding also highlights the importance of our paper.

We hope that the way we organize the introduction provides more clarity of what we did.

  1. Results are well presented, the article is well written, I have only a little concerns of the real impact of the using this cluster model on all cross sectional studies. 

Answer: Please note that we are not advocating for the use of cluster study designs in cross-sectional studies.

In the introduction we wrote “People in prisons share similar environmental characteristics (exposures) because they are naturally grouped (confined) within cells, courtyards and prisons. This is considered a natural cluster (Figure 1). Among observational studies, cross-sectional designs may have cluster structure mainly by 1) people are grouped in a natural way (prisons, households, clinics, medical practices and neighborhoods), and 2) by sampling design ranging from stratified random sampling to multi-stage sampling [19–21]. People who are grouped in a natural way tend to have more similar exposures regarding the event of interest, and therefore failing to consider this structure in the analyses leads to inappropriate statistical inferences [22–24].”

  1. I have no more comments, good work.

Answer: Thank you so much.

Sincerely,

Diana Marín
